# Factors Influencing Visual Acuity in Patients with Active Subfoveal Circumscribed Polypoidal Choroidal Vasculopathy and Changes in Imaging Parameters

**DOI:** 10.3390/diagnostics13183017

**Published:** 2023-09-21

**Authors:** Fan Xia, Peiyu Xing, Hao Zhang, Tongtong Niu, Qi Wang, Rui Hua

**Affiliations:** 1Department of Ophthalmology, The Fourth People’s Hospital of Shenyang, China Medical University, Shenyang 110001, China; lovemelovetheworld@163.com (F.X.); 13897985586@163.com (P.X.); laozhang678678@sina.com (H.Z.); tongtongniu968@hotmail.com (T.N.); stereofancy@163.com (Q.W.); 2Department of Ophthalmology, First Hospital of China Medical University, Shenyang 110001, China

**Keywords:** optical coherence tomography angiography, anti-VEGF, branch vascular network, polypoidal choroidal vasculopathy

## Abstract

We performed a retrospective, observational study of 51 eyes in 51 treatment-naïve patients with polypoidal choroidal vasculopathy (PCV), whose lesion ranged within the 6 × 6 mm scope of optical coherence tomography angiography (OCTA). The patients were divided into an ill-defined group and a well-defined group based on the pattern of branching vascular network (BVN) on OCTA. BVN morphology was not related to baseline best-corrected visual acuity (BCVA). However, the BCVA in the ill-defined BVN group (−0.18 [interquartile range: −0.40 to 0.00]) was significantly improved after anti-vascular endothelial growth factor (VEGF) injections, compared with that (0.00 [interquartile range: −0.18 to 0.00]) in the well-defined group (z = 2.143, *p* = 0.032). Multiple logistic regression analysis showed that male sex, fewer injections, and the presence of polypoidal lesions on OCTA images at baseline predicted a poor prognosis in patients with polypoidal lesions on OCTA images after anti-VEGF therapy (all *p* < 0.05). Finally, BCVA at baseline and the number of injections were protective factors for BCVA after anti-VEGF therapy (all *p* < 0.05). In contrast, a history of hypertension and macular edema at baseline were risk factors for BCVA after anti-VEGF injections (all *p* < 0.05). Our results revealed the visual and morphological prognosis of patients with active subfoveal circumscribed PCV after anti-VEGF therapy.

## 1. Introduction

Polypoidal choroidal vasculopathy (PCV), a subtype of neovascular age-related macular degeneration (nAMD), is characterized by polypoidal lesions at the terminus of the branching vascular network (BVN) on indocyanine green angiography (ICGA) [1]. The increasing use of multimodal imaging, especially optical coherence tomography (OCT) and OCT angiography (OCTA), has provided cumulative evidence that PCV is vascular and originates from the type 1 neovascular network [2]. Recently, PCV was classified into the pachychoroid disease spectrum, characterized by Haller’s layer vessel dilatation and attenuated choriocapillaris [3]. The relationship between these choroidal changes and the locations of BVN ingrowth suggests that pachychoroid features underlie the pathogenesis of PCV lesions [4]. To the best of our knowledge, although some studies examined the relationship between the morphological characteristics of polypoidal lesions and visual acuity, no study has examined its relationship with BVN. Therefore, this study investigated the relationship between BVN morphology and changes in visual acuity, as well as its associated imaging features. Given the variable size and morphology of PCV, cases with massive hemorrhage or BVN extending beyond the scope of OCTA may assume a course or nature that is different from that of cases with PCV confined to the macular area with limited hemorrhage. Therefore, we conducted our study on PCV cases characterized by limited hemorrhage confined to the macula.

Anti-vascular endothelial growth factor (VEGF) monotherapy is the preferred treatment to optimize visual outcomes for PCV, while minimizing the number of injections required. However, standard imaging parameters to determine visual and structural prognosis are still lacking. Our study aimed to analyze the effect of anti-VEGF therapy in patients with PCV, assess their morphological changes, and investigate the parameters that influence the prognosis of visual acuity.

## 2. Materials and Methods

### 2.1. Eligibility Criteria and Anti-VEGF Treatment

This was a retrospective study based on real-world evidence. A medical record review of patients who were first diagnosed with PCV using ICGA was performed from January 2019 to May 2022 at the Department of Ophthalmology, the Fourth People’s Hospital of Shenyang, China Medical University. The diagnosis of PCV was made by identifying early-phase subretinal ICGA hyperfluorescence spots that appeared within the first 5 min after ICG dye injection, [5] along with at least one of the following findings: (1) orange subretinal nodules; (2) branching vascular network supplying the polypoidal lesions; and (3) pulsatile filling of polyps. A polypoidal lesion in ICGA images was defined as an aneurysm or aneurysmal hyperfluorescence lesion within the first 5 min after ICG dye injection.

The inclusion criteria were as follows: (1) a diagnosis of treatment-naïve PCV based on the EVEREST criteria [5]; (2) confirmation of PCV activity by both leakage on fluorescein angiography (FA) and exudation on OCT [6]; (3) the presence of PCV lesions that were within the OCTA 6×6 macular examination range; (4) macular hemorrhage that did not impede BVN morphological recognition [7]; (5) a history of receiving three plus pro re nata (3 + PRN, PRN dosing with 3 initial monthly injections followed by PRN) [8] intravitreal injections of anti-VEGF treatment, with subsequent follow-up at 1 month after each injection; and (6) the absence of reactivation in the study eye for at least 6 months after the last injection. No reactivation meant no intraretinal or subretinal fluid (SRF) on OCT and no need for further anti-VEGF therapy or other treatment [9]. The exclusion criteria were as follows: (1) the presence of any other fundus diseases that could potentially confuse the results, such as retinal angiomatous proliferation, retinal arterial microaneurysms, and retinal artery or vein occlusion; (2) previous ophthalmic intervention (such as laser, vitrectomy, or photodynamic therapy); (3) refractive medium turbidity that could impede BVN morphological recognition or image discrimination; (4) patients with high pigment epithelium detachment (PED) that could affect the measurement of choroidal parameters; (5) quiescent PCV and PCV with massive hemorrhage (a quiescent PCV is characterized by the presence of one or several polyps in the absence of any intraretinal or subretinal fluid or hemorrhage and usually does not need treatment [10], while a PCV with massive hemorrhage is defined as the presence of four or more disc areas with subretinal or sub-RPE hemorrhaging [11]); and (6) patients with serious systemic diseases (such as heart or brain infarction, liver or kidney failure, infectious diseases, rheumatic diseases, blood diseases, etc.)

In total, 51 eyes in 51 patients satisfied all the inclusion and exclusion criteria (Figure 1). All patients received anti-VEGF intravitreal injections using ranibizumab, conbercept, or both. Medication alterations resulted from inadequate responses, defined as persistence of SRF on OCT after 3 sessions of the loading phase injection [12]. 

### 2.2. Clinical Measurements

All patients underwent examinations including best-corrected visual acuity (BCVA) in decimal form, fundus photography (TRC-50DX; Topcon, Tokyo, Japan), FA, ICGA (Spectralis HRA; Heidelberg Engineering, Heidelberg, Germany), enhanced-depth imaging OCT (Spectralis HRA + OCT; Heidelberg Engineering), and OCTA (RTVue AngioVue System, Optovue, Inc., Fremont, CA, USA) of a macular cube (6 × 6 mm), centered at the fovea. All data were measured by two masked retinal specialists (F.X. and P.X.), and the average value was used in the final evaluation. In case of disagreement, a third senior retinal specialist (H.R.) made the final judgment.

We recorded the PCV-related parameters including: late geographic hyperfluorescence (LGH) on ICGA; polypoidal lesions on OCT and OCTA (polypoidal lesions on OCT were considered sub-RPE ring-like structures in cross-sectional OCT [13], while polypoidal lesions on OCTA were recognized as tangled or coil-like structures on en face OCTA that corresponded to the roundish location on en face OCT and blood flow signal on the cross-sectional OCT within the same image [14]); BVN on ICGA and OCTA; double-layer sign (DLS) and pachyvessels on OCT; macular edema on OCT including intraretinal fluid and intraretinal cystoid regions; central foveal thickness (CFT); SRF height; SFCT; and PED height. According to the medication that was administered, all enrolled patients were categorized into three groups in the subgroup analysis: ranibizumab, conbercept, and combined groups, to analyze differences in their efficacy. According to OCTA flow maps, BVN was divided into four subtypes based on previous nAMD studies [15,16], including an ill-defined type, the medusa (or sea-fan shape) type, the mulberry type, and the dead tree type (Figure 2). Alterations among these subtypes were counted. Furthermore, the mulberry type and medusa (or sea-fan shape) type are more likely to be active, whereas the other two types are likely to be less active. As the dead tree type represents an advanced stage in the disease progression, which is fundamentally distinct from the other types, it was omitted from the subsequent comparisons. We divided the remaining patients into an ill-defined group and a well-defined group, the latter including patients with the mulberry and medusa (or sea-fan shape) types. Furthermore, the differences in the parameters between the two groups were compared. The medical history of hypertension was also documented.

### 2.3. Statistical Analyses

Statistical analyses were performed using SPSS (version 23.0; IBM Inc., Chicago, IL, USA). The data are presented as medians along with interquartile ranges. Categorical variables are expressed as numbers. Decimal BCVA was converted to the logMAR form for statistical analysis. Differences in BCVA between different anti-VEGF injection groups were analyzed using one-way ANOVA tests. Wilcoxon matched-pair signed-rank tests were used to investigate the differences between quantitative data and ranked data. The Spearman’s correlation coefficient was used to investigate factors that influence the prognosis of BCVA and polypoidal lesions on OCTA. Moreover, the factors that influence the prognosis of BCVA and polypoidal lesions on OCTA were assessed using linear regression analysis and multiple logistic regression analysis, respectively. Kappa analysis was used to value the interobserver agreement between the reports of the two examiners on the subtypes of BVN. The Wilcoxon–Mann–Whitney test was used to investigate differences in imaging parameters between the ill-defined and the well-defined groups. The gender distribution between these two groups was analyzed using the Chi-Square test. Odds ratios (ORs) and 95% confidence intervals (CIs) were also calculated. Statistical significance was defined as a *p*-value < 0.05.

## 3. Results

### 3.1. General Patient Information

After excluding patients with quiescent PCV and PCV with massive hemorrhage, we enrolled 51 PCV eyes (26 right eyes and 25 left eyes) in 51 patients with a median age of 65.0 years (interquartile range: 62.0–71.0 years), including 27 male and 24 female patients. The average number of injections was five (interquartile range: 4 to 8); we performed three injections in three eyes; four injections in 10 eyes; five injections in 17 eyes; six injections in four eyes; seven injections in two eyes; eight injections in five eyes; nine injections in four eyes; 10 injections in two eyes; 11 injections in two eyes; 12 injections in one eye; and 18 injections in one eye.

All the groups were compared at baseline. The BCVA was: ranibizumab group (0.70 [interquartile range: 0.655–1.325]), conbercept group (0.70 [interquartile range: 0.40–1.00]), and combined group (0.71 [interquartile range: 0.40–1.00], F = 0.355, *p* = 0.703), and there was no statistically significant difference between them. Compared with the baseline parameters (0.70 [interquartile range: 0.40–1.00]) of all eyes, the BCVA at the last visit (0.60 [interquartile range: 0.40–1.00]) had significantly improved (z = 3.093, *p* = 0.002). However, there was no difference in BCVA improvement between the ranibizumab (0.70 [interquartile range: 0.52–1.24]), conbercept (0.52 [interquartile range: 0.30–1.00]), and combined groups (0.56 [interquartile range: 0.375–0.82], F = 2.438, *p* = 0.098) at the last visit. There was also no change in the number of macular edemas on OCT. Furthermore, the shape of the BVN on OCTA changed in several patients, but this was not statistically significant (Table 1).

The final BCVA was positively correlated with the baseline BCVA (r = 0.682, *p* < 0.001), the history of hypertension (r = 0.288, *p* = 0.040), the presence of macular edema at baseline (r = 0.540, *p* < 0.001), CFT at baseline (r = 0.415, *p* = 0.002), PED height at baseline (r = 0.370, *p* = 0.007), the presence of macular edema after anti-VEGF injections (r = 0.350, *p* = 0.012), CFT after anti-VEGF injections (r = 0.364, *p* = 0.009), and PED height after anti-VEGF injections (r = 0.279, *p* = 0.048). After adjusting for age, a linear regression analysis revealed that baseline BCVA (B = 0.383, 95%CI [0.193–0.572], t = 4.087, *p* < 0.001) and the number of injections (B = −0.030, 95%CI [−0.058–−0.002], t = 2.156, *p* = 0.037) were protective factors for BCVA after anti-VEGF injections. In contrast, a history of hypertension (B = 0.222, 95%CI [0.068–0.375], t = 2.913, *p* = 0.0060) and the presence of macular edema at baseline (B = 0.297, 95%CI [0.040–0.555], t = 2.330, *p* = 0.025) were risk factors for BCVA after anti-VEGF injections.

### 3.2. Variation and Correlation of Imaging Parameters

#### 3.2.1. BVN Shape on OCTA

According to the OCTA profile at baseline, 15 eyes were regarded as ‘ill-defined’ type, 23 eyes were regarded as ‘mulberry’ type, 12 eyes were regarded as ‘medusa (or sea-fan shape)’ type, and one eye was regarded as ‘dead-tree’ type. This final classification was established by the senior retinal specialist (H.R.). There was a disagreement between the two examiners (F.X. and P.X.) regarding one image. The consistency of staging between the two examiners was verified using Kappa analysis, which yielded a Kappa value of 0.970, and was considered almost perfect agreement.

Four participants’ BVN morphology subtypes changed on OCTA after anti-VEGF treatment. One ill-defined type changed into the medusa type, one mulberry type changed into the medusa type, and two mulberry type changed into the dead tree type. However, the ratios of these changes were generally statistically insignificant (z = 0.855, *p* = 0.393).

Fifteen eyes were assigned to the ill-defined group, while 35 eyes in the mulberry type and medusa (or sea-fan shape) type were assigned to the well-defined group. The BCVA in the ill-defined group (−0.18 [interquartile range: −0.40 to 0.00]) significantly improved after the anti-VEGF injections, compared with that in the well-defined group (0.00 [interquartile range: −0.15 to 0.00]) (z = 2.143, *p* = 0.032). The improvement of CFT in the ill-defined group (211.00 [interquartile range: 73.00 to 305.00] μm) was more significant than that in the well-defined group (68.00 [interquartile range: −14.00 to 189.00] μm) (z = 2.371, *p* = 0.018). However, no difference was found in age, gender, number of injections, baseline BCVA, baseline SFCT, baseline CFT, SRF, baseline PED height, SFCT improvement, SRF improvement, and PED height improvement between the two groups (Table 2).

#### 3.2.2. Polypoidal Lesions on OCTA

In all 51 enrolled eyes, the polypoidal lesions on OCTA images after anti-VEGF injections were positively correlated with male sex (r = 0.367, *p* = 0.008), the presence of polypoidal lesions on OCTA images (r = 0.331, *p* = 0.018) at baseline, and the presence of polypoidal lesions on OCT images (r = 0.478, *p* < 0.001) after anti-VEGF injections. After adjusting for age, multiple logistic regression analysis showed that male sex (*p* = 0.005), the presence of polypoidal lesions on OCTA at baseline (*p* = 0.008), and a smaller number of injections (*p* = 0.033) predicted a poor prognosis for the polypoidal lesions on OCTA after anti-VEGF injections (Table 3).

#### 3.2.3. Changes in OCT-Related Parameters

In all 51 enrolled eyes, after anti-VEGF therapy, the CFT decreased significantly from 431.0 (interquartile range: 341.0–698.0) µm at baseline to 350.0 (interquartile range: 242.0–580.0) µm (z = 3.923, *p* < 0.001) at the last follow-up visit. There was also a reduction in SRF from 140.0 (interquartile range: 0.0–257.0) µm at baseline to 0.0 (interquartile range: 0.0–165) µm (z = 3.137, *p* = 0.002). Furthermore, the proportion of polypoidal lesions on OCT was 51.0% after the anti-VEGF injections, which was significantly lower than that at baseline (z = 2.183, *p* = 0.029). A reduction in pachyvessels on OCT was also observed from the baseline (49/51) to last visit (40/51, z = 3.0, *p* = 0.003). 

### 3.3. Comparison of the Incidence of Imaging Parameters at Baseline

In the early stages of ICGA, 68.6% of the eyes with PCV showed BVN, which was lower than the detection rate of the DLS on OCT (96.1%, z = 3.5, *p* < 0.001). Similarly, 60.8% of the eyes with PCV showed LGH in the late stage of ICGA, which was lower than the detection rate of choroidal pachyvessels on OCT (96.1%, z = 4.025, *p* < 0.001). For the non-invasive detection of polypoidal lesions, OCTA had a relatively lower detection rate (20/51) than OCT (35/51, z = 3.441, *p* = 0.001). OCTA demonstrated a comparable detection capability for BVN (70.6%) when compared to ICGA (68.6%, z = 0.243, *p* = 0.808).

## 4. Discussion

### 4.1. Factors That Influence the Prognosis of BCVA

Our study qualitatively and quantitatively analyzed the structural changes in 51 eyes in 51 patients with PCV before and after treatment using multimodal imaging. There was no difference in BCVA improvement between the different anti-VEGF injection groups, which is consistent with the findings of a previous study [17]. A significant trend of improving BCVA was observed compared to baseline; further, the final BCVA was related to the baseline BCVA, which is consistent with the findings of the LUMINOUS study [18]. The final BCVA was also related to a history of hypertension at baseline, similar to the report by Gupta et al. [19]. Furthermore, pre- and post-treatment macular edema, CFT, and PED height were also identified as factors that could affect the final BCVA. Similarly, Schmidt-Erfurth et al. indicated that progressive visual loss occurred in patients with PED and was induced by secondary macular edema in the neurosensory retina [20]. Yamamoto et al. also suggested that macular edema was associated with severe vision loss in eyes with end-stage PCV [21]. Anti-VEGF therapy is known to reduce neovascularization and intra- and sub- retinal exudation. Therefore, persistent macular edema and higher CFT and PED could lead to a poorer recovery of vision. 

Furthermore, the ill-defined BVN group had a better improvement of vision than the well-defined group. Similarly, Cheung et al. reported that the area of BVN and the age were associated with changes in BCVA [22]. However, our study’s findings are slightly different, as no age correlation was identified. Furthermore, in this study, the BVN morphological subtypes were altered in several cases. The alteration of the BVN morphological subtypes was thought to be arterialized with thicker and dilated vessel trunks. The absence of tiny branches led to the transformation of an immature pattern into a mature pattern. The well-defined types present mostly mature dilated vessels and immature lesions which were significantly associated with choroidal neovascularization (CNV) growth [16], whereas the ill-defined type lacks this structure, which is full of tiny vessels. So, it is reasonable to assume that the well-defined type was possibly more mature or had a longer course than the ill-defined type, which resulted in more damage and poorer recovery of vision. To a certain extent, the visual prognosis of PCV can be predicted by considering CNV morphological subtypes. It is important to evaluate polypoidal lesions and BVN together when assessing PCV activity.

### 4.2. Variation and Correlation of Imaging Parameters

On OCT imaging, SRF height, CFT, polypoidal lesions, and pachyvessels were decreased significantly after treatment, which is similar to the reports of previous studies [1,23,24]. The regression of polypoidal lesions is a significant anatomical target for the treatment of PCV, as it may have implications for long-term disease stability [25]. Pachyvessels are representative signs of choroidal vascular hyperpermeability [26]. The presence of hyperpermeability is a significant risk factor for the development of active PCV [27]. The VEGF-related changes in the diameter of pachyvessels may regulate choroidal thickness and exudative changes; furthermore, the diameter of pachyvessels may be a biomarker of exudative change [28].

On OCTA imaging, the presence of pre-treatment polypoidal lesions on OCTA, fewer injections, and male sex predicted a poor prognosis of the polypoidal lesions on OCTA after treatment. A study by Liu et al. supports our observations. It reported that male patients with the rs17030 G allele in complement component 3 gene had an increased risk of PCV [29]. Polypoidal lesion regression as a treatment goal for PCV management remains controversial. Recent clinical trials showed improved visual acuity with inactive but persistent polypoidal lesions [30]. Chang et al. indicated that polypoidal lesions detected by OCTA were more likely to be persistent than those with a negative flow signal on OCTA [30].

### 4.3. Comparison of the Incidence of Different Imaging Parameters

Comparing the incidence of baseline imaging parameters revealed that the detection rate of the DLS on OCT was higher than that of BVN on ICGA. In our study, the incidence of BVN (68.6%) was similar to that reported in a previous study [23]. The term BVN is commonly used to describe the whole network area or the relatively large vessels in PCV eyes [24], which is now considered type 1 macular neovascularization [30]. Previous histology studies showed that the space within the DLS contained macular neovascularization as well as serous fluid, exudation, and thickened extracellular matrix material or was thickened beneath the basal laminar deposits [24,29]. A previous study indicated that the extent of the DLS on OCT was larger than the BVN on ICGA in some cases [18]. Therefore, the detection probability of the DLS was greater than that of BVN, although the DLS often spatially represents the area of the BVN [17], even though they are not identical.

We found that the detection rate of pachyvessels on OCT was higher than that of LGH on ICGA, although LGH and pachyvessels are thought to be related to choroidal vascular hyperpermeability. LGH was defined as a well-demarcated geographic hyperfluorescent lesion in the late phase of ICGA [31]. Our results are similar to those of a previous study in which LGH was noted in most PCV eyes [32]. Kim et al. found that all eyes that developed active PCV had LGH at baseline and vice versa [32]. Vessel dilation may derive from vortex venous engorgement and anastomosis, and macular choroidal vessels may not modulate the pressure by vortex vein outflow; hence, the regional vessel pressure could be abnormally high and may result in vascular remodeling [31]. Therefore, pachyvessels may be an early change that appears before the clinical manifestation of PCV, which coincides with the reports in a study by Siedlecki et al. [33].

We assessed two non-invasive examinations to detect polypoidal lesions and found OCTA had a relatively lower detection rate than OCT. Our study showed similar detection rates of polypoidal lesions as those of previous studies on OCT and OCTA [34,35]. Ring-like or bubble-like changes on OCT are highly suggestive of polypoidal lesions and are characteristic of PCV. Although slow and turbulent flow within polypoidal lesions or circulation at the periphery of an aneurysmal dilation might contribute to poor detection on OCTA [36], these cannot fully replace ICGA. In this regard, the Asia Pacific Ocular Imaging Society PCV workgroup recently provided recommendations for the non-invasive diagnosis of PCV [37].

This study has some limitations. This was a retrospective, single-center study with a limited sample size. Patients with large hemorrhage or PEDs were excluded, so that all parameters could be compared in the same individual; therefore, the findings primarily refer to PCV eyes with limited hemorrhage. Furthermore, we not have a size criterion for pachyvessels; thus, we just conducted a qualitative analysis based on their appearance on OCT. We abandoned the analysis of choroidal vascular density because its measurements were subjected to significant interference by artifacts and hemorrhage. All these limitations should be considered in future research.

## 5. Conclusions

In conclusion, our results revealed the visual and morphological changes in patients with active subfoveal circumscribed PCV. The baseline BCVA and the number of injections were protective factors for the final BCVA, whereas a history of hypertension and macular edema were risk factors. This study categorized the shape of the BVN. Patients with PCV with ill-defined BVN patterns had better improvement in vision. This study also highlighted that male sex, a smaller number of injections, and the presence of polypoidal lesions on OCTA images at baseline were risk factors for the prognosis of polypoidal lesions on OCTA images.

## Figures and Tables

**Figure 1 diagnostics-13-03017-f001:**
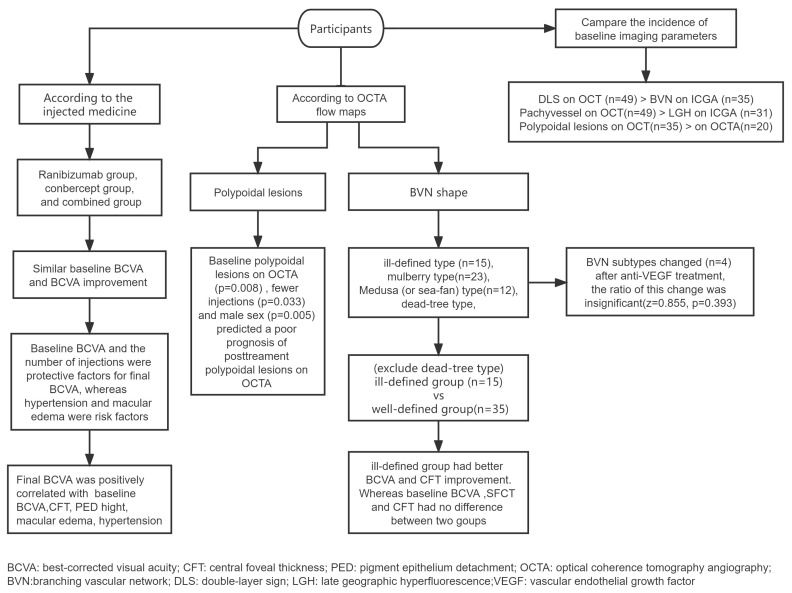
Experimental procedure.

**Figure 2 diagnostics-13-03017-f002:**
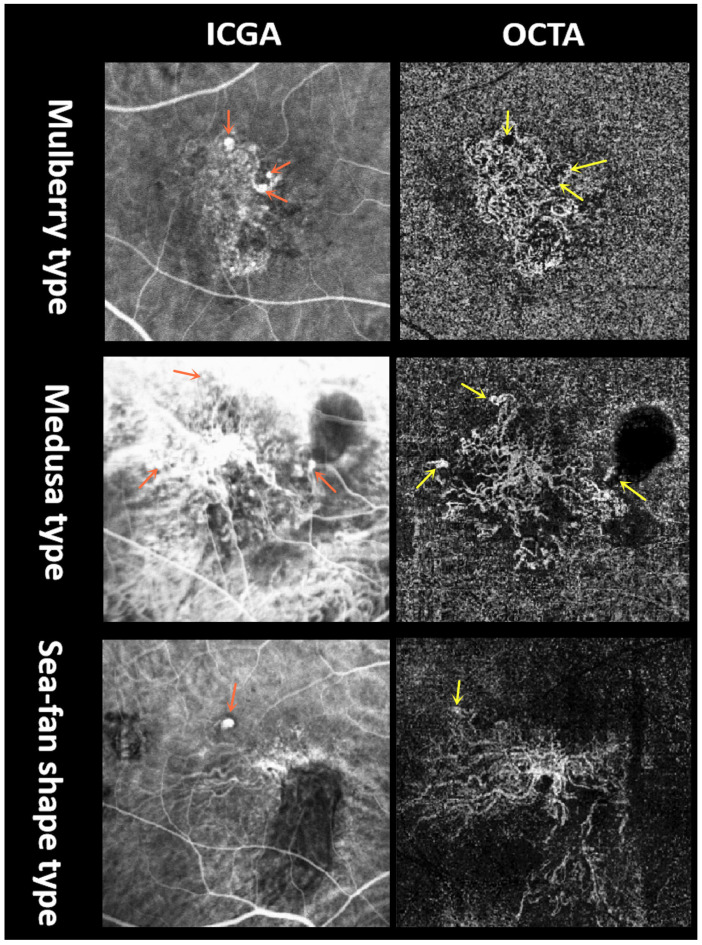
Well-defined subtypes of BVN on OCTA and corresponding ICGA images. ICGA revealed the polypoidal lesions (red arrows), which were also identified by OCTA (yellow arrows).

**Table 1 diagnostics-13-03017-t001:** General information and measurements.

Variable	Baseline	Last Follow-Up	*p* Value
LogMAR BCVA	0.70 (interquartile range: 0.40–1.00)	0.60 (interquartile range: 0.40–1.00)	z = 3.093, *p* = 0.002 *
Macular edema (OCT)	12/51	12/51	z = 0.000, *p* = 1.000
BVN (OCTA)	Ill-defined type, 15 eyes	Ill-defined type, 14 eyes	
	Mulberry type, 23 eyes	Mulberry type, 20 eyes	z = 0.855, *p* = 0.393
	Medusa or sea-fan shape type, 12 eyes	Medusa or sea-fan shape type, 14 eyes	
	Dead tree type, 1 eye	Dead tree type, 3 eyes	

* *p* < 0.05.

**Table 2 diagnostics-13-03017-t002:** Comparison of imaging parameters between the ill-defined and the well-defined groups.

Variable	Ill-Defined Group (*n* = 15)	Well-Defined Group (*n* = 35)	*p*
Age, years	66.00 [interquartile range: 62.00 to 73.00]	64.00 [interquartile range: 61.00 to 70.00]	0.379
Gender, male/female	Male = 6/9	Male = 21/14	0.193
The number of injections (n)	5 [interquartile range: 4 to 7]	5 [interquartile range: 4 to9]	0.522
Baseline BCVA, logMAR	0.70 [interquartile range: 0.30 to 1.22]	0.70 [interquartile range: 0.52 to 1.00]	0.907
Baseline SFCT, μm	356.00 [interquartile range: 187.00 to 389.00]	259.00 [interquartile range: 191.00 to 360.00]	0.244
Baseline CFT, μm	446.00 [interquartile range: 376.00 to 722.00]	424.00 [interquartile range: 307.00 to 598.00]	0.295
Baseline SRF, μm	128.00 [interquartile range: 0.00 to 297.00]	145.00 [interquartile range: 0.00 to 257.00]	0.923
Baseline PED height, μm	41.00 [interquartile range: 0.00 to 114.00]	63.00 [interquartile range: 0.00 to 252.00]	0.418
BCVA improvement, logMAR	−0.18 [interquartile range: −0.40 to 0.00]	0.00 [interquartile range: −0.15 to 0.00]	0.032 *
SFCT improvement, μm	21.00 [interquartile range: −26.00 to 80.00]	4.00 [interquartile range: −34.00 to 38.00]	0.295
CFT improvement, μm	211.00 [interquartile range: 73.00 to 305.00]	68.00 [interquartile range: −14.00 to 189.00]	0.018 *
SRF improvement, μm	84.00 [interquartile range: 0.00 to 259.00]	0.00 [interquartile range: 0.00 to 110.00]	0.087
PED height improvement, μm	0.00 [interquartile range: 0.00 to 108.00]	0.00 [interquartile range: 0.00 to 32.00]	0.500

*p* Value was calculated by Chi-Square (gender) and Wilcoxon–Mann–Whitney tests (other parameters), * *p* < 0.05.

**Table 3 diagnostics-13-03017-t003:** Factors that influence the prognosis of polyps on OCTA.

	B	OR	95%CI	*p*
Age	0.050	1.051	0.962–1.148	0.274
Sex	2.577	13.153	2.187–79.100	0.005 *
PLs on OCTA at baseline	2.364	10.637	1.840–61.485	0.008 *
The number of injections	−0.412	0.663	0.454–0.967	0.033 *

PLs: polypoidal lesions; * *p* < 0.05.

## Data Availability

The data presented in this study are available on reasonable request from the corresponding author and with appropriate permissions from the Fourth People’s Hospital of Shenyang. The data are not publicly available due to ethical reasons.

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
