# Peer review of "Factors Influencing Visual Acuity in Patients with Active Subfoveal Circumscribed Polypoidal Choroidal Vasculopathy and Changes in Imaging Parameters"

_diagnostics, 2023, doi:10.3390/diagnostics13183017_

Round 1
Reviewer 1 Report
The study is well written and brings new information of ophthalmological interest related to retinal pathology.
It includes a small group of patients, it would be interesting to follow these changes on a larger groups.
The number of anti-VEGF intravitreal injections is not uniform on the lot. Is any difference between the number of injections and the obtained results?
Reviewer 2 Report
Dear authors,
It is an interesting paper in which are studied several factors influencing visual acuity in patients with active polypoidal choroidal vasculopathy (PCV) primarily reported a new correlation between BCVA and the Branch Vascular Network (BVN) of Polypoidal Choroidal Vasculopathy (PCV).
This study provides a further contribution in the field of PCV. The main reported finding is the possible correlation between BCVA and the type of BVN morphology.
Nevertheless, the reviewer recommends to the authors an explanation about the statistical analysis concerning the calculation of the level of significance (p value). Analyzing multiple comparison, it should be set a lower level of significance (p<0,01), which would some data no more significant statistically.
Minor changes.
Line 140: should be specified you are talking about all groups together, I suppose.
Line 142: should be specified “statistically difference”. In fact, there are some small differences between groups.
Line 111: medical history of hypertension has been recorded. Although, this parameter is positively correlated with BCVA. Is it possible that history of hypertension affects branch Vascular Network?
In the abstract and in the conclusion is remarked “circumscribed” PCV, why not to add this specification to the title? It would be more specific.
I would reorganize the presentation of the results. In paragraph 3.3 it says that OCTA and OCT detected only 20 ad 51 PCV at the baseline, respectively. For this reason, I suppose that results displayed in Table 3 are referred not to all 51 eyes but only on this aspect. Please clarify this point
Reviewer 3 Report
a significant research on a highly important but less understood topic.
a well written article
Reviewer 4 Report
This is a well-constructed research focusing on predictive factors for VA of the eyes with PCV.
Specific comments:
1) BVN was divided into four subtypes: ill-defined type, Medusa (or sea-fan shape) type, mulberry type, and dead tree type. Because this classification is based on the appearance of OCTA images, the authors should describe how many examiners looked at 1 image and the discrepancy of staging between examiners.
2) Table 3. The group identified polyps using ICGA images. However, according to Figure 2, presences and sizes of polypoidal lesions are not clear. Please describe how they defined polyps in ICGA images.
3) Results 3.3. The group compared detection rates of polyps between OCT and OCTA. They should mention how they defined polyps in OCT images.
None.
Round 2
Reviewer 4 Report
The authors reponded very well.
None.